# Meeting the Physical Activity Recommendations and Its Relationship with Obesity-Related Parameters, Physical Fitness, Screen Time, and Mediterranean Diet in Schoolchildren

**DOI:** 10.3390/children7120263

**Published:** 2020-11-28

**Authors:** José Francisco López-Gil, Javier Brazo-Sayavera, Wagner de Campos, Juan Luis Yuste Lucas

**Affiliations:** 1Departamento de Actividad Física y Deporte, Facultad de Ciencias del Deporte, Universidad de Murcia (UM), 30720 San Javier, Spain; 2Department of Sports and Computer Science, Universidad Pablo de Olavide (UPO), 41013 Seville, Spain; jbsayavera@cur.edu.uy; 3Polo de Desarrollo Universitario EFISAL, Centro Universitario Regional Noreste, Universidad de la República (UDELAR), 40000 Rivera, Uruguay; 4Centro de Estudos em Atividade Física e Saúde CEAFS, Universidade Federal do Paraná, 81531-980 Curitiba, Brazil; wagner-campos@hotmail.com; 5Departamento de Expresión Plástica, Musical y Dinámica, Facultad de Educación, Universidad de Murcia, 30100 Espinardo, Spain; jlyuste@um.es

**Keywords:** obesity, sedentary behaviour, feeding patterns, overweight, healthy lifestyle, children

## Abstract

The up-to-date scientific evidence suggests that adequate levels of physical activity provide essential health benefits for children and adolescents and help to maintain a healthy body weight. In this sense, children and adolescents should at least accumulate 60 min of moderate-to-vigorous physical activity in a daily basis to achieve these benefits and be considered active. Likewise, some lifestyle-related elements may interact with each other in an antagonistic or synergistic way to modify physical activity status. Thus, a better understanding of how meeting physical activity recommendations influences these potentially modifiable lifestyle factors (obesity-related parameters, physical fitness, dietary habits, or sedentary behaviour) would significantly reinforce the importance of complying with those recommendations from a health perspective and support the establishment of strategies for the promotion of diminishing the lower trends of physical activity among the young population. This study seeks to verify the association of meeting physical activity international recommendations with obesity-related parameters, global physical fitness, screen time, and Mediterranean diet in Spanish schoolchildren aged 8 to 13. A cross-sectional study was performed including 250 schoolchildren (41.2% girls) aged 8–13 (9.7 ± 1.2) from six primary schools in the Region of Murcia (Spain). Results: A higher proportion of children who complying with physical activity recommendations shows normal weight, no abdominal obesity, and low adiposity in comparison to other with different obesity-related parameters categories. Higher values in global physical fitness score were found in those who meet the physical activity international recommendations in both sexes. These higher values were also shown for adherence to the Mediterranean diet in both sexes; not being so in the case of screen time. Notwithstanding, none of these mean differences were statistically significant. To conclude, the proportion of schoolchildren meeting the physical activity recommendations in our study is low. A higher proportion of children who meet with physical activity recommendations present normal weight, no abdominal obesity and low adiposity in comparison to other obesity-related parameters categories in both sexes. Likewise, those considered as active children seem to have higher global physical fitness score and adherence to the Mediterranean diet than children who do not meet the recommendations.

## 1. Introduction

The up-to-date scientific evidence suggests that adequate levels of physical activity (PA) provide essential health benefits for children and adolescents, and help to maintain a healthy body weight [1,2]. In this regard, children and adolescents should at least accumulate 60 min of moderate-to-vigorous physical activity (MVPA) in a daily basis to achieve optimal health benefits [1,3], since PA represents the most variable element of total energy expenditure [4]. Notwithstanding, a large proportion of youth do not meet these recommendations [5,6]; with only 30–40% of youths being sufficiently active (evaluated by self-report) as determined by public health recommendations [7,8]. As far as Spain is concerned, it has recently been pointed out that 63.3% of the Spanish child and adolescent population (8 to 16 years old) does not meet the World Health Organization (WHO) recommendation of daily MVPA [9]. In addition, the Spanish report card on PA indicators in the Global Matrix 3.0 informed of rather insufficient levels in PA level of children and indicated the limitation of periodic data on PA level in young population [10]. Thus, information about this parameter, as well as on the modifiable factors that might influence it, is therefore needed to implement efficient public policies designed to improve the levels of PA among population to attain the 2025 worldwide aim of decreasing the levels of insufficient PA [11].

On the other hand, low levels of PA [12] and a sedentary lifestyle [13] have been associated with obesity in both children and adolescents; reaching the global prevalence of excess of weight is a worrying trend in the youngest population [14]. Similarly, the Region of Murcia has been pointed out as the region of Spain with the highest prevalence of overweight/obesity [15]. In this line, meeting the MVPA recommendation has been the recommendation most strongly linked to obesity, in relation to other healthy recommendations (e.g., sleep time and screen time) when these recommendations have been analysed in isolation [16]. Regarding physical fitness (PF), the scientific evidence available on secular trends suggests a decreasing in physical fitness of children and adolescents [17]. About this worrying situation, previous systematic reviews have well defined its positive association with certain cardiometabolic risk indicators in children and youth [18,19]; especially in those who practice 60 or more minutes of MVPA daily [2]. Additionally, a higher prevalence of children not meeting has been reported in Spain [9]. In this sense, it has been suggested that screen time is linked to the displacement of time available for PA [20]. In terms of eating habits, Mediterranean Diet (MD) is classified among the healthiest eating patterns in the world [21] given its specific properties, the type of food consumed, as well as an active lifestyle [22]. However, a worrying tendency towards the abandonment of the MD in children and adolescents has been indicated [23,24]. On this concern, the most of the studies performed show a positive relationship between MD adherence and PA [23].

In support of this notion, some of these lifestyle-related elements may interact with each other in an antagonistic or synergistic way to modify PA status [25]. For instance, a large amount of screen time (as a way of sedentary behaviour) can result in a change in the hours spent in PA, increasing the energy consumption from eating while watching and/or reducing sleep hours [13]. Likewise, children who meet the PA recommendations could eat more because of their higher energy expenditure to maintain an energetic balance [26]. Thus, a better understanding of how meeting of PA recommendations influences on these potentially modifiable lifestyle factors (obesity-related parameters, PF, dietary habits or sedentary behaviour) would significantly reinforce the importance of complying with that recommendations from a health perspective and support the establishment of strategies for the promotion of diminishing the lower trends of PA among the young population. Likewise, this fact is more essential during childhood, since in this age phase begins the process of cardiovascular diseases as well as their associated risk factors; increasing the risk of premature mortality [27].

Hence, according to the evidence and the lack of information regarding the compliance of PA recommendations in Spain and especially in the Region of Murcia, as well as the importance of further learning about meeting PA recommendations and the previously mentioned lifestyle-related factors, the aim of this research was to determine the proportion of schoolchildren meeting the PA recommendations and, then, to compare obesity-related parameters, global PF, screen time and MD according to the meeting of PA recommendations in Spanish schoolchildren aged 8 to 13.

## 2. Materials and Methods

### 2.1. Sample

All schools in the five different municipalities that compose the Valley of Ricote (Region of Murcia, Spain) were invited to participate in this cross-sectional study. Thus, six primary schools agreed to participate. There were 250 schoolchildren (41.2% females) aged 8–13 (9.7 ± 1.2) from six primary schools with similar socio-demographic characteristics. To this end, participants were chosen using a non-probability sampling.

Children had to be authorised by their parents or legal guardians in order to participate in the study. Both parents and their children received prior information concerning the objective of the research and the type of testing that would be conducted. As a criterion for exclusion in the study, those children who were exempt from participation in Physical Education classes were not included.

This study was undertaken in conformity with the Helsinki Declaration, with full respect for the human rights of the children and was ratified by the Bioethics Committee of the University of Murcia (ID 2218/2018).

### 2.2. Data Collection

#### 2.2.1. Sociodemographic Data

Participants’ age and sex were declared by themselves. The type of schooling was divided into 2 categories: public and private with public funding. Additionally, area of residence was dichotomised into urban (>5000 inhabitants) and rural (≤5000 inhabitants).

#### 2.2.2. Physical Activity

Participants fulfilled the Physical Activity Questionnaire for Older Children (PAQ-C) to provide an estimation of the moderate-to-vigorous physical activity they engaged in the previous seven days [28]. This tool has been validated and translated into Spanish [29] and has nine items which are rated on a 5-point Likert scale. For further analysis, the percentage of minutes per day of MVPA was calculated according to Saint-Maurice et al.’s equation [30]. Thus, two categories were established: ‘meeting PA recommendations’ (≥420 min per week) and ‘not meeting PA recommendations’ (<420 min weekly), according to the international recommendation of 60 min of daily MVPA (on average) [1].

#### 2.2.3. Anthropometric Measurements

Participants’ height was measured by means of a portable height rod with a measuring precision of 0.1 cm (Leicester Tanita HR 001, Tokyo, Japan). Their body weight with a digital scale (with a precision of 0.1 kg) (Tanita BC-545, Tokyo, Japan). A constant tension tape with an accuracy of 0.1 cm was used to determine the waist circumference, at the point between the edge of the iliac crest and the last rib. Waist-to-height ratio was calculated dividing the waist circumference by the height (both in cm). Skinfolds were measured with a precision of 0.2 mm with professional steel plicometers (Holtain, Crosswell, Crymych, UK) at the triceps, biceps, iliac crest, and subscapular. The Siri formula was applied to estimate body fat mass from body density [31].

#### 2.2.4. Obesity Related-Parameters

Participants’ body mass index (BMI) was determined by dividing weight in kilograms by height in meters squared. Considering the WHO age- and sex-specific thresholds, BMI z-score was computed, and then weight status was determined [32]. Abdominal obesity was established by the waist-to-height ratio cut off point of 0.5 [33]. Likewise, body fat percentages higher than 25% for boys and 30% for girls were considered as a ‘high adiposity’ [34].

#### 2.2.5. Physical Fitness

PF was assessed by the extended Assessing Levels of Physical Activity (ALPHA) health-related fitness test battery, previously described in detail and validated in children and adolescents [35]. This battery includes certain tests in order to determine cardiorespiratory fitness (20-m Shuttle Run Test), upper muscular strength (handgrip strength test), lower muscular strength (standing broad jump) and motor fitness (4 × 10 Shuttle Run Test). To generate an estimation of the global PF variable, the results of the four different tests were transformed into z-scores by sex and age as follows: z-score value = (value − mean)/standard deviation (SD). Then, the sum of these z-scores was computed to establish the global PF score [36]. All participants completed the test during their Physical Education lessons.

The very same team of researchers carried out all the measurements. Such measurements were taken as a circuit, in which participants completed each test one after another, except for the cardiorespiratory fitness test, which was performed by several participants at the same time. The physical measurements took one hour per school group to be completed.

#### 2.2.6. Screen Time

Children were questioned about the average time they normally spent in both daily screen time (TV or videogames) with the next question: ‘‘On an average school day, how many hours do you watch TV or play videogames?’’. Screen time was categorised into “low screen time” (less than 2 h per day) and “high screen time” (2 h per day or more), according to the Canadian recommendations for screen time [25].

#### 2.2.7. Mediterranean Diet

In order to determine the adherence to the MD, the Mediterranean Diet Quality Index for children and adolescents (KIDMED) index was used [37]. It consists of 16 questions with an index ranging from 0 to 12. Items that imply negative connotations with respect to MD score as −1, and those with positive connotations with +1. In this way, by adding up the results, three different levels are obtained: (1) >8, high MD; (2) 4–7, moderate MD; (3) ≤3, low MD [37].

### 2.3. Data Management and Statistical Analysis

All the continuous variables were presented as a mean (M) and standard deviation (SD), and the categorical variables as a number (*n*) and percentage (%). The assumption of the normality was checked by Kolmogorov–Smirnov test with Lilliefors’ adjustment. To verify the homogeneity of variances, Levene’s test was applied. Consequently, the data were examined using Mann–Whitney U test or Student’s *t* test for two-groups comparisons; based on whether or not the assumption of normality is met. In the case of finding differences between groups, post-hoc tests were carried out by Mann–Whitney U test with Bonferroni adjustment. Contrariwise, for normally distributed variables, Dunnett T3 test or Tukey’s honestly-significant-difference (HSD) was applied according to the homogeneity of the variances. Relationships between categorical variables were estimated by chi-square test. Analyses of covariance (ANCOVA) were applied to assess the mean differences of the dependent variables (obesity-related parameters, global PF score, screen time hours, and KIDMED score) with respect to the participants meeting PA recommendations (‘meeting PA recommendations’ and ‘not meeting PA recommendations’), after adjusting by several covariates. Prior to this, variables that had biased distributions were transformed. To help interpretation, the two-step approach to transform normally undistributed continuous variables into normal distributions was used [38]. The included covariates were age, type of schooling (public/private) and area of residence (rural/urban). Data analysis was carried out by the software SPSS (IBM Corp., Armonk, NY, USA) for Windows (v. 24.0). The statistical significance was set at *p* < 0.05.

## 3. Results

Data of age, anthropometry, PF tests, MD, screen time, and sport activities of the sample, according to sex are shown in Table 1. Girls had higher body fat percentage than boys; however, no statistically significant differences were found according to obesity-related parameters between sexes. In the case of PA, boys showed higher scores than girls in PAQ-C as well as the number of minutes of weekly MVPA. Likewise, a higher proportion of boys met the PA recommendations (54.4%); showing statistically significant difference compared to girls (12.6%). Conversely, girls scored higher than boys in KIDMED test.

Figure 1 shows the proportion of participants meeting the MVPA recommendations and the different independent variables of the study. In both sexes, a higher proportion of participants who met the PA recommendations were of normal weight, no abdominal obesity, and low adiposity compared with children with different obesity-related parameters categories. Likewise, most of the girls meeting the PA recommendations presented medium or high global fitness score (84.6% for the sum of both categories). Additionally, a higher proportion of children that met the PA recommendations also comply with the Canadian recommendations of screen time; especially in the case of girls (69.2%). However, no statistically significant association was found for any variable. In the case of MD, only a low number of children considered as active showed high adherence to the MD (11.3% for boys and 7.7% for girls). Further information about frequencies (absolute and relative) of these associations are shown in the Appendix A.

Lastly, Table 2 states the mean differences of the independent variables of the study with respect to the compliance of PA recommendations, after adjusting by several covariates. Thus, the percentage of girls who did not meet the PA recommendations is higher (87.4%) than that of boys (45.6%); for the whole sample, 62.8%. According to obesity-related parameters; girls meeting the PA recommendations show lower values (BMI z-score, waist circumference, waist-to-height ratio and body fat percentage); not being so in the case of boys. Moreover, ANCOVA revealed higher values in global PF in those who meet the PA recommendations in both boys and girls. Likewise, these greater results were shown for MD in both sexes; not being so in the case of screen time. Notwithstanding, none of these mean differences were statistically significant.

## 4. Discussion

The aim of this study was to determine the number of schoolchildren complying the PA recommendations and to compare obesity-related parameters, global PF, screen time, and MD in relation to the meeting of PA in Spanish schoolchildren aged 8–13.

In line with the results obtained, girls show less minutes of weekly MVPA, as well as a lower proportion of participants meeting the PA recommendations. In the whole analysed sample, it was noted that six out of every ten children are considered as insufficiently active, with higher prevalence in the case of girls. These results are in line with those obtained in other international recent studies [5], as well as others in Spain [9,10] or in the Region of Murcia [39]. This low prevalence of active children emphasises the need to establish public policies that provide effective interventions in families, schools and communities in order to increase PA levels in young population [5].

On the other hand, not meeting the MVPA recommendation has been linked to a greater association with childhood obesity regardless of screen time and sleeping time [16,40]. In this line, we found a greater proportion of children with normal weight, no abdominal obesity and low adiposity who met the PA recommendations in both boys and girls. However, a lack of statistical significance was found in our sample. A possible justification for this discrepancy could be justified by the different tools for measuring the PA level [41]. Additionally, this fact could be partially explained due to the high prevalence of children who have overweight/obesity in the Region of Murcia; it being the region with the most obese children [15]. Another reason for this lack of association could be the fact that MVPA only accounts for a small proportion (<5%) of the 24-h period, with the remaining 95% dedicated to sleep, sedentary behaviour, and light-intensity activities [42]. In that concern, some studies indicate that most children spend a large proportion of their awake time being active at low levels of intensity [43,44,45], especially girls [45]. Notwithstanding, other study only found weak associations between the time spent in light-intensity activities and BMI z-scores [16].

In addition to the above, children who meet the PA recommendations presented higher scores of global PF, especially in the case of girls. In this concern, substantial data indicated that health and fitness benefits will occur in most children and youth who participate in 60 or more minutes of MVPA on a daily basis [2]. However, within groups with similar fitness, the risk is higher in insufficient active individuals compared with those who comply PA recommendations [27]. Although much of the variability in physical condition is genetically determined, environmental determinants and especially PA influence PF [46].

The excessive time spent in sedentary behaviours, and particularly screen time, has been negatively associated with health outcomes in the scientific literature [42]. The odds of engaging in daily PA decreased significantly with increasing time spent in daily screen time, for both sexes, and greater volumes of daily screen time were associated with lower odds of daily PA practice for both boys and girls [47]. Additionally, another study shown higher association between meeting the PA recommendations and the screen time international recommendations for both sexes [48]. In this line, we found that most of the children considered active comply with Canadian recommendations of screen time (less than 2 h per day), mainly in the case of girls. However, we did not find statistically significant differences between the compliance of the PA recommendations and the number of hours spent in screen time in our sample. Thus, the main reason could be the fact that we inquired how many hours they spent watching TV or playing videogames, without examining the use of smartphones, tablets, and other small screens currently used by children and youngsters [42]. Moreover, another reason that could justify this lack of association is the higher amount of screen time shown in our sample. Thus, more than half of our sample do not comply with the Canadian recommendations proposed by Tremblay et al. [25]. These results coincide with the high prevalence that it has been recently found in Spanish children and adolescents during the week (more than 50%); almost 80% being those who not meet with the recommendations on weekends [9].

With regards to the adherence to the MD, only a small proportion of children considered as active showed a high adherence to the MD. This fact could be explained by the low proportion of children who met the PA recommendations in our study (about one in three participants) and the low number of participants with high MD (one in four children). This low proportion of children with high adherence to the MD is in line with the tendency towards the abandonment of the MD, especially in youths [24]. Nevertheless, we found higher mean values of KIDMED score were shown for those who meeting the PA recommendations, especially in girls. These results agree with the previous findings of Iaccarino et al. [23] who reported a positive relationship between greater levels of PA and KIDMED score in their systematic review. At the same time, these results match with those offered by Tambalis et al. [48] in both boys and girls, as well as those indicated by other authors [49,50]. However, these authors employed a different instrument to determine the PA level, which could have an influence on the results. On the other hand, the fact of being physically active and less sedentary and the preference of taking healthy food has been reported in a previous study [51]; the fact of following healthy eating habits (e.g., MD) is an important marker towards embracing more active behaviour. Concerning the justification of this association, earlier studies, mainly performed in adolescents, suggest that those who practise more PA could desire a superior performance, choosing healthier meals to attain it [52]. Additionally, there is the fact that more PA needs more energy expenditure [4] and those who report higher PA would be expected to have a greater consumption of essential nutrients [53] than the MD might provide.

The main strength of this study is the wide combination of factors studied (obesity-related parameters, global PF, adherence to the MD and screen time) in relation to the compliance of PA recommendations. Likewise, another strong point is the addition to current knowledge in Spanish children, in order to satisfy the need indicated on the lack of information on PA indicators [10]. Conversely, in this research, we found certain limitations. Firstly, given the cross-sectional nature of this research, it cannot be concluded that the associations found indicate causal association. Additionally, the sample was not recruited using a random selection (probability sampling). However, despite this, all schoolchildren from the sampled schools were encouraged to participate. Another limitation that we found is related to the lack of information on socioeconomic status, sleeping hours, or individual’s developmental stage. For example, Tanner Stage, as an indicator of sexual maturation, may have provided a more accurate figure of cardiorespiratory fitness, as evidence has been found of its use in monitoring fitness tests [54]. Moreover, if we had measured PA objectively using accelerometer devices, it would have given us a more precise calculation of PA patterns and sedentary behaviour. Notwithstanding, the model offered by Saint-Maurice et al. was highly correlated with the recorded accelerometer values (*r* = 0.63).

Meanwhile, current PA recommendations from WHO establish that youth should at least accumulate 60 min of MVPA daily; most of it being aerobic and vigorous intensity activities. Nevertheless, other international recommendations indicate that it would be recommended to change the statement “at least 60 min of MVPA per day” to “at least 60 min of MVPA per day on average” [25]. The formula that we have used to estimate the minutes of MVPA, calculates the total amount of weekly minutes from the percentage of daily minutes of MVPA; not being able to verify that children exceed every day the 60 min of MVPA, as indicated by the WHO. Consequently, there are currently no specific recommendations for light-intensity activities; this is necessary in future research for a better understanding of all the components of movement in the 24-h period on different health outcomes [16]. For this reason, collective and international collaborative efforts are needed to modify the persistent trends of low PA and high screen time among children and adolescents around the world [5].

## 5. Conclusions

To conclude, the proportion of schoolchildren meeting the PA recommendations in our study is low. When comparing children, with respect to meeting international PA recommendations, higher proportions of those who meet with PA recommendations present normal weight, no abdominal obesity, and low adiposity in comparison to other obesity-related parameters categories in both sexes. Likewise, those considered as active children seem to have higher global PF score and adherence to the MD than children who do not meet the recommendations. Nevertheless, this association was not found in the case of the hours of screen time.

From a health perspective, these results emphasise the need to promote policies and interventions aimed at encouraging the practice of PA among school children, in order to improve their health through increased PF. These policies should be targeted especially at girls, to ensure participation in sport activities that ensure a minimum of PA practice and reduce sedentary habits.

## Figures and Tables

**Figure 1 children-07-00263-f001:**
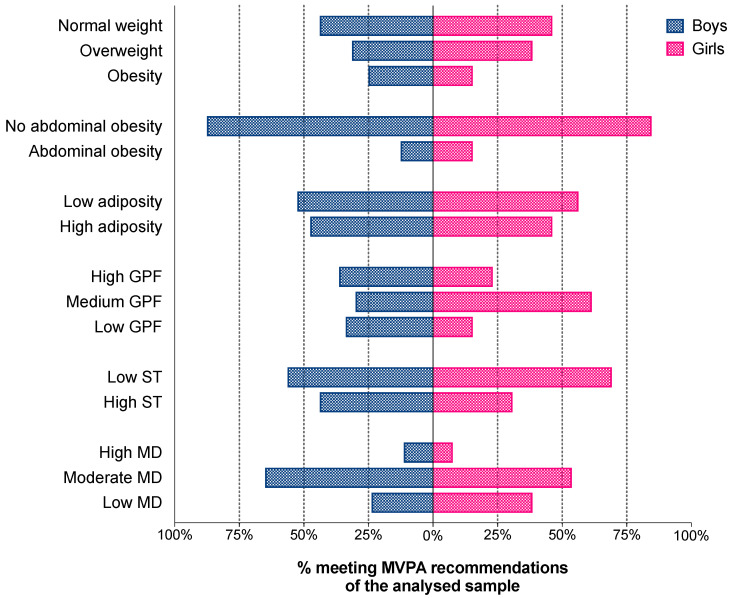
Proportion of participants meeting the moderate-to-vigorous physical activity and according the different dependent variables of study. Data expressed as relative frequencies. GPF: Global physical fitness; MVPA: Moderate-to-vigorous physical activity; MD: Mediterranean Diet; ST: Screen time; WHtR: Waist-to-height ratio. Adjusted by age, type of schooling and area of residence.

**Table 1 children-07-00263-t001:** Descriptive data of the sample stratified by sex.

Variables	Boys (*n* = 147; 58.8%)	Girls (*n* = 103; 41.2%)	*p*
Age (years)	9.7 (1.2)	9.8 (1.1)	0.937
Weight (kg)	39.70 (10.33)	39.56 (10.68)	0.984
Height (m)	1.42 (0.08)	1.41 (0.10)	0.479
BMI (z-score)	1.14 (1.23)	1.11 (1.14)	0.842
Overweight/Obese (%)	79 (53.7)	50 (48.5)	0.418
WHtR [WC (cm)/Height (cm)]	0.46 (0.04)	0.45 (0.06)	0.095
Abdominal obesity (%)	20 (13.6)	18 (17.5)	0.402
BF (%)	30.03 (6.76)	32.25 (7.67)	0.033
High adiposity (%)	74 (50.3)	58 (56.3)	0.352
Global PF (z-score)	−0.01 (2.95)	0.00 (3.09)	0.997
Daily screen time (hours)	1.6 (1.1)	1.5 (0.9)	0.561
Daily screen time < 2 h (%) ^a^	78 (53.1)	60 (58.3)	0.417
KIDMED (score)	5.8 (2.1)	6.3 (2.1)	0.091
High MD (%) ^b^	33 (22.4)	28 (27.2)	0.391
PAQ-C (score)	2.27 (0.53)	2.13 (0.41)	0.038
Weekly MVPA (min)	422.6 (60.5)	361.5 (51.0)	<0.001
Weekly MVPA ≥ 420 min (%) ^c^	80 (54.4)	13 (12.6)	<0.001

Data presented as mean (standard deviation). BF: Body fat; BMI: Body mass index; KIDMED: Mediterranean diet quality Index for children and teenagers; MD: Mediterranean diet; MVPA: Moderate-to-vigorous physical activity; PAQ-C: Physical Activity Questionnaire for Older Children; PF: Physical fitness; WC: Waist circumference; WHtR: Waist-to-height ratio. ^a^ Proportion of participants who meet the screen time recommendations (<2 h) [25]. ^b^ Proportion of participants with high adherence to MD. ^c^ Proportion of participants who meet the physical activity recommendation (≥420 min) [1].

**Table 2 children-07-00263-t002:** Mean differences of obesity-related parameters, global physical fitness score, screen time hours, and KIDMED score, according to the meeting of physical activity recommendations and sex.

Variables	Boys			Girls		
	MPAR (*n* = 80; 54.4%)	NMPAR (*n* = 67; 45.6%)	*p ^#^*	MPAR (*n* = 13; 12.6%)	NMPAR (*n* = 90; 87.4%)	*p ^#^*
BMI (z-score)	1.37 (0.17)	0.95 (0.19)	0.147	0.63 (0.37)	1.06 (0.12)	0.277
WHtR (WC/Height(cm))	0.46 (0.01)	0.46 (0.01)	0.634	0.44 (0.02)	0.45 (0.01)	0.659
BF (%)	30.74 (0.93)	29.20 (1.05)	0.347	29.70 (2.40)	32.62 (0.82)	0.266
Global PF (z-score)	0.06 (0.43)	−0.07 (0.49)	0.850	0.72 (0.95)	−0.12 (0.33)	0.370
Daily screen time (hours)	1.6 (0.2)	1.6 (0.2)	0.909	1.7 (0.3)	1.5 (0.1)	0.583
KIDMED (score)	6.2 (0.3)	5.5 (0.3)	0.181	7.4 (0.6)	6.1 (0.2)	0.082

Data expressed as mean (standard error). BF: Body fat; BMI: Body mass index; KIDMED: Mediterranean diet quality Index for children and teenagers; MPAR: Meeting physical activity recommendations; NMPAR: Not meeting physical activity recommendations; PF: Physical fitness; WHtR: Waist-to-height ratio. ^#^ Adjusted by age, type of schooling and area of residence.

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
