# Peer review of "Meeting the Physical Activity Recommendations and Its Relationship with Obesity-Related Parameters, Physical Fitness, Screen Time, and Mediterranean Diet in Schoolchildren"

_children, 2020, doi:10.3390/children7120263_

Round 1
Reviewer 1 Report
A couple of very minor pickups in the editing:
156 (obesity-related parameters, global PF score, screen time hours and KIDMED score) with respect to the
158 recommendations’), after adjusting by several covariates.
Table 1 – p values should be consistent ie 0.xxx (see BF%)
Reviewer 2 Report
Thank you for trying to address my comments. However, this new version still has room for improvement. Please see following for my additional suggestions
Introduction
1) By looking the title “meeting the Physical activity recommendations, and its relationship with obesity-related parameters, physical fitness, screen time and Mediterranean diet in school children”, we are expecting two things
1a. meeting physical activity recommendation
1b. its relationship with obesity-related parameters, physical fitness, screen time and Mediterranean diet in school children
However, none of these were well justified in introduction.
- Authors explained the importance of meeting physical activity recommendation; however, it is unclear what is the research gap and why study that in school children, etc.
- It is remaining unclear why study the relationship between meeting physical activity recommendation and obesity-related parameters, physical fitness, screen time and Mediterranean diet in school children. Authors did state the importance of physical fitness, screen time. However, it is unclear why study the relationship between meeting physical activity recommendation and them, no research gaps have been identified.
- Obesity-related parameters was not justified in introduction
2) The logic flow in introduction remains my concern. My suggestions are to
- Justify the importance of meeting physical activity recommendation by identify research gaps from existing literature.
- Justify why it is needed to study the relationship between meeting physical activity recommendation and those identified parameters (e.g., obesity-related parameters, physical fitness, screen time and Mediterranean diet) by identifying research gaps in existing literature.
Method
1) Obesity-related parameters should be specified in a particular section. Although authors mentioned height, weight, waist to height ratio etc. it is unclear whether they are obesity-related parameter.
Discussion
1) It is unclear to me why authors did not discuss the relationship between meeting physical activity recommendation and “obesity-related parameters”
2) page 11, lines 228 -239, it is unclear why this paragraph aimed for since it is neither appear to state the strength nor weakness of the present study.
3) page 11, lines 240-245, the conclusion is not really consistent with results.
Round 2
Reviewer 2 Report
My previous comments have not been well addressed in this version. Introduction remain my major concern. Please see my comments below
1) Introduction still not well justified and lacking of logic flow. It is still unclear what is the existing research gaps regarding meeting physical activity recommendation’s relationship with obesity-related parameters, physical fitness, screen time and Mediterranean diet in school children
For example, authors made some following statements which only indicated that these relationships have been well established. However, it is unclear what are the existing research gaps and why study these relationships are meaningful.
1) page 2, lines 66-68 “similarly, meeting the MVPA recommendation in an isolate way has been the recommendation most strongly linked to obesity, among others (e.g., seep time and screen time).
2) page 2, lines 74-75 “ it is not surprising that most of the studies performed show a positive relationship between MD adherence and PA”
ALSO,
As I mentioned in my pervious comment “By looking the title “meeting the Physical activity recommendations, and its relationship with obesity-related parameters, physical fitness, screen time and Mediterranean diet in school children”, we are expecting two things: 1a). meeting physical activity recommendation; 1b). its relationship with obesity-related parameters, physical fitness, screen time and Mediterranean diet in school children. However, IF authors only present one part (1b). They should update their topic to just focusing on meeting physical activity recommendations’ relationships with those parameters.
2) Conclusion still not really conclude the present study (page 9, lines 328-333)
Your title is “meeting the Physical activity recommendations, and its relationship with obesity-related parameters, physical fitness, screen time and Mediterranean diet in school children”, whereas A) study aim was (page 2, lines 90-91): “ to verify the association of meeting MVPA recommendation with obesity-related parameters, global PF, screen time and MD in Spanish school children aged 8-13. B) conclusion was (page 9, lines 328-333) ”higher proportion of participants with normal weight, no abdominal obesity and low adiposity met PA recommendation”. It appears that there are inconsistency.
Author Response
Please see the attachment.

This manuscript is a resubmission of an earlier submission. The following is a list of the peer review reports and author responses from that submission.
Round 1
Reviewer 1 Report
Thank you for the opportunity to review this manuscript titled “Meeting physical activity guidelines and its relation with physical fitness, dietary habits and screen time in schoolchildren”. Although there are lots of strength about this manuscript, there are still room for improvement. My major concern is that whatever written in the manuscript is not well align with manuscript title, and study aim specified in abstract and introduction. Here are my suggestions for authors to consider
Title
1) please make sure the title of the manuscript reflects the focus of the manuscript. For example, authors used “Mediterranean Diet” and “dietary habits” interchangeably but they are different.
Abstract
1) Please make sure that study results align with study aim
For example, the study aim is about whether meeting physical activity guidelines made difference for physical fitness, Mediterranean Diet and screen time. Whereas, the results indicated that (page 1, lines 25-26) “a positive statistically significant correlation was found between the weekly minutes of moderate-vigorous physical activity and physical fitness… …” This is confusing whether the focus is meeting physical activity guidelines or physical activity.
2) Please make sure that all abbreviations are appropriately defined, such as PF, KIDMED etc.
3) authors used “Mediterranean Diet” and “dietary habits” interchangeably but they are different.
4) Please report statistics in the results whenever appropriate.
5) It would be great to have a brief conclusion
Introduction
1) Please justifying study using children related research instead of adults (page 1, lines 35-36; page 2, line 47, line 50).
2) Whenever abbreviations first appear, it should be appropriately defined. Please check the entire manuscript and make sure that is the case. For example, WHO appears in introduction (page 2, line 61) but no defined until page 3, line 107. There are others such as ALPHA etc.
3) There are two separate things about physical activity in introduction: physical activity guidelines and physical activity level. Authors need to address this, either adjust study aim to reflect whatever written in introduction OR modify introduction to align with study aim
4) Also, it appeared to me based on information provided that the present study examined the relationship between meeting physical activity guidelines and screen time whereas author justified the relationship between screen time and sleep (page 2, lines 48-49). This is especially confusing to me when it comes to the independent variables and dependent variables. Thus, it would be helpful if authors make it clear in their study aim(s). Meanwhile, please make sure that introduction is written in such a way to set the stage for your study.
Method
1) it is fine to use convenient sampling but it should be indicated as one of study limitations in discussion.
2)physical fitness measure is lacking of information, what is it, how it conducted, scored, and its criteria etc. Especially there are multiple fitness tests, e.g., shuttle run, hand-grip etc. Is there a total fitness score encompass all these tests? How authors considered students fitness results while drawing conclusion on fitness in general based on these fitness test results.
3) In table notes, analytic methods should be included since table itself should be stand alone.
4) Table 1, Table 2 and Table 4 look fine to me. Meanwhile, I would like to suggest authors explain fitness measure better in method. For example, since it is from fitness test battery, it should have a composite score or a total score for all individual fitness tests included and this total score should be reported in results and Table 2 and Table 4
5) I don’t understand what is the purpose of Table 3 since the study aim is focused on “meeting physical activity guideline” instead of moderate and vigorous physical activity time. Again, authors should make their study aim clear, then whatever described in abstract, introduction, method and results should align with that.
Discussion
Authors explained “The Mediterranean Diet quality Index for children and teenagers (KIDMED)” well in method and its score – overall and three classifications. And reported in results. However, whatever discussed in discussion (pages 13, lines 256-258, page 14, lines 259-266) are not aligned with or related to that. Please consider rewrite and reorganized your discussion regarding to KIDMED and make sure your discussion reflects your results. Also, dietary habits and Mediterranean diet used interchangeably, please correct that and make sure it is consistent throughout your manuscript.
Reviewer 2 Report
Dear authors,
I have several comments about your paper. It is a pleasure for me to contribute to your study.
- I suggest you improve the first sentence of your abstract.
- I recommend you rewrite the study design. Is it a descriptive study? I suggest you change to associative. For example, an associative and cross-sectional study, because of the main results is from an association test.
- Please, could you give us my information about how was the sample selected? How have you selected the schools?
- In the abstract section please could explain what is the meaning of PF.
- You have shown several results that not are included in your main purpose. I suggest you keep your manuscript more coherent considering the purpose, methods, and results. In this way I suggest you: Purpose: 1) to verify the relationship between meet and not meet physical activity guidelines with physical fitness, body composition, diet habits, and physical activity after school in children. However, here we have some problems: if you decided to work with a categorial analyses maybe you need to have a great sample. In this case, I suggest you include a sample size calculation for the statistical test that you decided to use. It is very important to highlight the potential of your results.
- I suggest you rewrite the statics analyses considering the description of which were the purpose of each statistical test. In my opinion, you have done several analyses to answer the same research question. Thus, I recommend you keep for example only the regression analyses and after that, as I can see in your results, maybe you can try to do a mediation or moderation analysis by BMI/%fat or age. Because all your significative results seem to be dependent on these variables. So in this case, your analyses of the relationship could be better explained by a mediation or moderation approach.
- Table 3: It is not clear what you have done on this table. Please could you review it? It is a relationship between the PA and the other variables? So, I do not know what the differences between the results present in table 2 and table 3. I suggest you keep just the table 3 and after that test the moderation role of BMI/BF% etc. in the relationship between physical activity and fitness.
- Again: please I suggest you turn this table more auto explanate. It was very hard to understand these results. For me, you have done different test analyses to answer the same research problem. Maybe you need to review what is your main purpose.
- On the discussion topic, there are several results that you have discussed that I am not found in your results. Please I suggest you take out.
- Line 226 -233: I agree, however, these results are not in agreement with all published in the literature. Considering all components of 24 hours MVPA have been appointed as a variable with more strong association. Why did you not explore the continuum relationship between MVPA, PF, and the mediation and moderation role of both BMI, PF%, and age?
- Line 237: Here I agree. PF is a condition and in fact, has been showing a strong relationship with body composition and genetics determinants.
- 261: Or maybe the health diet habits could be a cause of an adequate level of PA.
- Finally, I suggest you present a more profound discussion about your results. You need to highlight the most important results that you have. For example, why did you not included a discussion about the role of PF%, age, and BMI in our results?
Thank you.
Regards
Round 2
Reviewer 1 Report
The manuscript is certainly improved. Thank you for taking my suggestions into consideration. However, there are still room for improvement. I strong suggest authors to proof read the manuscript before resubmission.
Abstract
- the brief background (page 1, lines 17-19) is good to have however, it should be able to justify the purpose of the present study. However, I did not see diet or screen time. However, I don’t understand how “maintain a healthy body weight” have anything to do with the present study.
- page 1, line 21 “an associative and cross-sectional study” what does this means. I understand cross-sectional study but add “an associative” only cause more confusion.
- Abbreviation still not fully explained such as KIDMED
- Information provided still not fully consistent with study aims/purpose and It still hard to follow sometimes, such as “ (page 1, line 28) children considered as active” which was not defined, another example is (page 1, line 27 and line 29), “sport activities hours” especially when the study is only focused on met/not met physical activity recommendation as authors mentioned for this updated manuscript
- Screen time was part of study focuses but not reported in results
Introduction
- research question still not well justified (page 2, lines 50-52, lines 53-58, lines 66-68). For example,
Example 1: authors utilized “maintain a healthy body weight” which is neither part of study aims/purposes nor measured and explained in study method. Also, this background still doesn’t explain well why it is essential to study the relationship between meeting physical activity recommendation with physical fitness and Mediterranean Diet.
Example 2: authors mentioned to examine the relationship between PA level and body composition (page 2, line 67). However, body composition was not studied in the present study
Two major concerns in the introduction a) only using necessary information to justify the research question, b) logic organization
- minor: some abbreviation still not defined such as KIDMED
Method
- whatever instruments used should aligned with study purpose and should cite resource and score criteria. For example, page 3, lines 100-101, it is unclear to me why “weekly sport activities” examined, Is it a dependent variable? Also, what “an ad hoc questionnaire”, what is it, what is the score criteria, resources should also be cited.
- also, instrument used for data collection should be better organized, normally, people report main instruments used for independent (e.g., Meeting PA guidelines) and dependent variables (e.g., screen time etc.), then demographics. Instead of independent variable, demographics, dependent variables etc.
- Although more information for fitness test was provided, it is still unclear the score criteria (it should be similar to what you did for diet [page 3, lines 125-126])
Discussion
Discussion was much better organized and easier to follow. One additional suggestion, why KIDMED was mentioned in method and conclusions but not in discussion (lines 235-246).
